# Predicting Future Cognitive Decline from Long-term Observations of Dual-task Performance Data

Shuqiong Wu
*Institute of Scientific and Industrial Research (SANKEN)*
*Osaka University*
Osaka, Japan
wu@am.sanken.osaka-u.ac.jp

Tomoya Noguchi
*Institute of Scientific and Industrial Research (SANKEN)*
*Osaka University*
Osaka, Japan
noguchit@am.sanken.osaka-u.ac.jp

Fumio Okura
*Graduate School of Information Science and Technology*
*Osaka University*
Osaka, Japan
okura@ist.osaka-u.ac.jp

Yasushi Yagi
*Institute of Scientific and Industrial Research (SANKEN)*
*Osaka University*
Osaka, Japan
yagi@am.sanken.osaka-u.ac.jp

*Abstract*—Early stage detection of cognitive decline is crucial for effective prevention and treatment of dementia. However, current approaches based on MRI or biomarkers are expensive and impractical, making them unsuitable for early-stage detection from daily measurements. A suitable option is the dual-task paradigm, which involves simultaneously performing two tasks (typically a physical task combined with a cognitive task). This approach has proven effective in assessing daily cognitive status. The underlying principle is that dual-task performance reflects the maximum cognitive load that can be handled by participants, which in turn reflects their current cognitive function. However, a one-time dual-task test cannot predict future changes in cognitive function. In this study, we present the first attempt at leveraging long-term observations of dual-task performance data. Our results show that changes in dual-task performance over time are associated with future cognitive changes. Our approach extracts temporal features from six months of dual-task performance data, and predicts future cognitive decline over the next two years using a machine learning model. Our experimental results yielded an accuracy comparable to that returned by MRI scans, thus demonstrating that the proposed approach can achieve early detection of future cognitive decline from routine dual-task measurements.

*Index Terms*—Dual-task, Dementia, MMSE, Early-stage detection, Mild cognitive impairment

## I. INTRODUCTION

With aging populations, dementia has emerged as a serious issue in our society. Dementia is a collection of diseases characterized by the progressive deterioration of memory and mental abilities, ultimately leading to severe disability in daily life. Dementia remains incurable with current medical treatments. However, early detection and intervention can impede progression of this disease, especially when identified at or before the mild cognitive impairment (MCI) stage, which is considered an early phase of dementia. Consequently, early detection of cognitive impairment is critical for preventing dementia among the elderly [1].

This work was supported in part by the Japan Agency for Medical Research and Development (AMED) under Grant JP24uk1024001.

Detecting early-stage cognitive impairment poses several challenges, because the initial symptoms are subtle and easily overlooked by the affected individuals. Paper-based examinations such as the Mini-Mental State Examination (MMSE), Montreal Cognitive Assessment (MoCA), Alzheimer's Disease Assessment Scale-Cognitive Subscale, HDS-R (Hasegawa's dementia scale, revised), are commonly employed in clinics for screening cognitive impairment [2]–[6]. However, the reliability of these screening tools for early-stage detection is hampered by the practice effect (PE), wherein individuals consistently score higher than their true cognitive function as a result of frequent assessments [7].

The dual-task paradigm represents an effective alternative to the above methods. This approach involves performing two tasks simultaneously (typically a physical task combined with a cognitive task), and has been widely adopted to detect early-stage cognitive impairment [8]–[15]. Dual-task-based approaches have proven effective in detecting early-stage cognitive impairment. However, a one-time dual-task test cannot predict future changes in cognition: the information provided by short-term dual-task assessments is insufficient for predicting the longitudinal changes in cognitive function.

To address the aforementioned issues, we propose a novel approach for predicting future cognitive decline that leverages long-term observations of dual-task performance data. To the best of our knowledge, our study represents the first attempt at exploiting long-term behavioral cues for predicting future cognitive function. Our experimental results demonstrate that there is indeed a correlation between future cognitive decline and changes in preceding sequential dual-task performance data. The two main contributions of our research are:

(1) a novel method for predicting future cognitive decline based on long-term observations of dual-task performance data. Unlike existing methods that rely on MRI scanning and biomarkers, our research represents a first attempt at exploring the correlation between preceding long-term be-

havioral cues and future cognitive changes, thereby offering valuable insights into cognition trends and potential precursors of cognitive decline at significantly earlier stages. Furthermore, the proposed system can be applied to long-term cognitive monitoring. In our study, it was implemented for both cognitive monitoring and data collection in multiple elderly centers.

(2) a database of long-term observations of dual-task performance data paired with MMSE scores over time. This database includes data from 39 participants collected over approximately 5 years, and is essential for evaluating the effectiveness of the proposed method in predicting future cognitive decline. This dataset holds significant value for neurologists and researchers, facilitating comprehensive analysis of cognitive function over extended periods.

## II. RELATED WORK

In this section, we present an overview of the latest advancements in cognitive assessment technology, alongside recent research related to predicting future cognitive decline.

### A. Cognitive assessment technology

As mentioned previously, paper-based assessments remain the primary approach for screening cognitive impairment in clinical settings [2]. For instance, MMSE is a commonly used assessment tool. It consists of a 30-point questionnaire that necessitates administration by trained medical professionals, typically on a one-on-one basis. It contains questions related to language ability, visual skills, memory, orientation to time and place, attention, and calculation [3]. Scores between 24 and 27 indicate a relatively high risk of MCI, while scores below 23 indicate a high probability of dementia. Paper-based assessments are effective and cost-efficient, however they are not ideal for frequent measurements because of their vulnerability to PE. Additionally, the scores returned by these assessments may vary depending on the evaluator, resulting in inconsistent performance [3].

To address the above issues, behavior-based methods, such as dual-task assessments, have garnered significant attention from researchers. For example, Mancioppi et al. devised a novel dual-task paradigm for MCI detection involving two types of dual-tasks: FTAP (fore-finger tapping with cognitive task) and TTHP (toe-tapping with cognitive task). Despite its novel features, this approach involves fixed cognitive questions that remain vulnerable to PE. Furthermore, it requires wearable sensors that may cause discomfort to the participants [8], [9]. Similarly, Digo et al. introduced a dual-task for screening cognitive impairment based on human gait features computed from wearable devices, which is not suitable for long-term monitoring [11]. Recently, Lillian et al. developed a new dual-task system involving three phases of walking combined with a cognitive task, such as counting backwards in 3's from 100 or reciting the alphabet. This approach has achieved a 81.97% sensitivity and 67.74% specificity (the sum was 1.4971) [12]. In 2021, Wu et al. were the first to apply STGCN [16] to dual-task-based cognitive impairment assessment [13]. They improved overall performance (with the sum of sensitivity and specificity reaching 1.76) by using spatio-temporal features from gaits. Subsequently, Liu et al. improved upon Wu et al.'s work and achieved a value of $\approx 1.90$ for the sum of sensitivity and specificity [14]. Godo et al. further enhanced the generalization ability of Liu et al.'s model by leveraging the periodicity of human gaits [15].

Although these peer-reviewed studies have demonstrated high performance in detecting current cognitive impairment, they are incapable of predicting future cognitive decline. The latter is a significantly more challenging task that cannot be addressed by a one-time behavioral assessment.

### B. Prediction of future cognitive decline

There are several studies that predict future cognitive decline from MRI data or biomarkers [18]–[23]. Current theories posit that many types of mental illness, including cognitive impairment, may arise from pathological changes. For example Alzheimer's disease, a typical form of dementia, is associated with brain changes such as brain atrophy and loss of neurons and synapses. Therefore, MRI data and biomarkers may reflect future cognitive decline caused by pathological diseases [1]. More recently, CSF (cerebrospinal fluid) has proven an effective biomarker for predicting future cognition [18]–[20]. Meanwhile, research based on MRI or PET (Positron Emission Tomography) data have achieved future prediction of cognitive impairment by detecting subtle pathological changes in brain structure [21]–[23]. However, MRI, PET, and biomarker tests are expensive and require medical supervision, making them unsuitable for frequent assessments.

In summary, previous approaches based on either dual-task behavioral data or clinical data (MRI, PET, or other biomarkers) cannot timely predict future cognitive decline, either as a consequence of insufficient information, or because of their unsuitability for frequent measurements. In this study, we exploit long-term behavioral cues to predict future cognitive changes. This approach involves daily-oriented measurements, which enable us to detect the earliest signs of behavioral changes related to future cognitive decline, even before MCI.

## III. METHOD

In this section, we provide a detailed explanation of the proposed architecture (depicted in Fig. 1) for detecting future cognitive decline. We define the problem at hand, describe the data collection system based on dual-task measurements, detail the feature extraction algorithm, and demonstrate our method for predicting future cognitive decline.

### A. Problem definition

In this study, we define the problem as predicting cognitive decline in the $\beta$ years following a baseline time $T_{\text{baseline}}$, as shown in the right part of Fig. 1. Baseline time is intended to indicate the current timestamp. To achieve this goal, we used a fixed period $\alpha$ months preceding the baseline time (left part of Fig. 1) as input data. Specifically, for each sample $i$, we define the baseline time point $T_{\text{baseline},i}$. Dual-task data measured within the interval $[T_{\text{baseline},i} - \alpha, T_{\text{baseline},i}]$ is represented as a

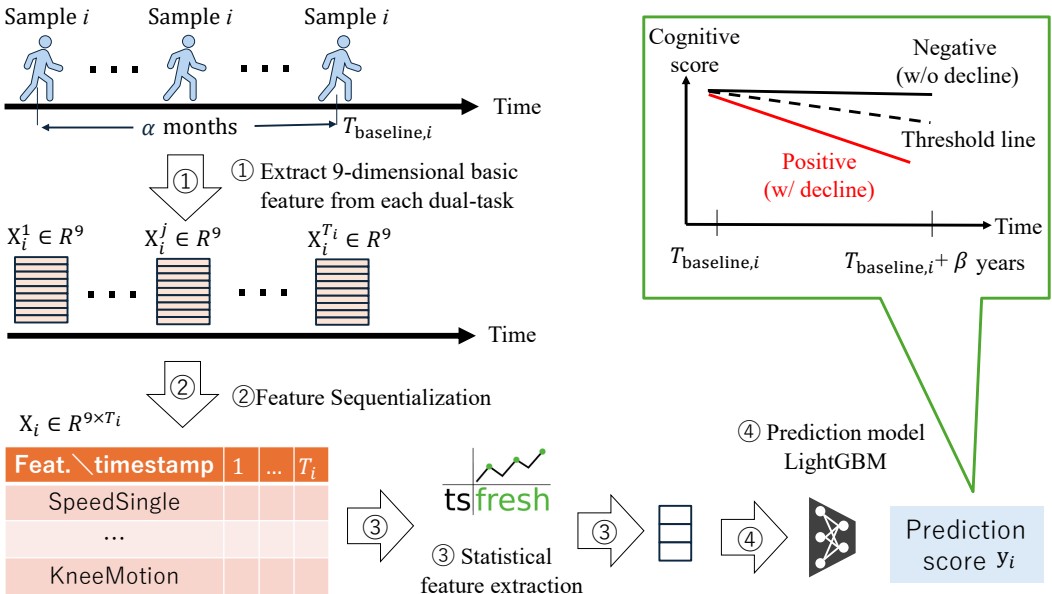

Fig. 1. Architecture of the proposed approach for detecting future cognitive decline

multivariate time series $\mathbf{X}_i \in \mathbb{R}^{C \times T_i}$, where $T_i \in \mathbb{N}$ represents the number of dual-task measurements conducted by sample $i$ within $\alpha$, and $C$ denotes the number of dimensions of the dual-task data for each measurement.

The label indicating whether cognitive function declines within the interval $(T_{\text{baseline},i}, T_{\text{baseline},i} + \beta]$ is denoted as $y_i \in \{0, 1\}$. The determination of label $y_i$ is based on overtime MMSE scores, where 0 indicates cognitive maintenance and 1 indicates cognitive decline. The reason for using MMSE scores over time to compute the ground truth data is that MMSE is one of the most widely used cognitive assessment tools worldwide. Since our proposed architecture is independent of the tool used to assess cognitive ability, MMSE can be replaced with other assessments, such as MoCA or HDS-R.

In summary, the problem setting of this study is to predict the final label $y_i$ for sample $i$ using the long-term dual-task observations $\mathbf{X}_i$ as input.

### B. Data collection with a dual-task system

In this study, we adopted a dual-task experience system developed by Okura et al. [24], and extended its use to obtain long-term measurements of dual-task performance. This system, as shown in Fig. 2, combines a physical task (stepping) with a cognitive task (arithmetic question). It includes a 30-second single calculation task, a 20-second single stepping task, and a 30-second dual-task (calculation while stepping). Performance in single-tasks separately is used as reference for dual-task performance, to mitigate the impact of inter-individual variability. The arithmetic questions are generated randomly to address potential issues associated with practice effects, making this approach suitable for frequent measurements [24]. During the dual-task test, a depth camera records movements during both single-task and dual-task stepping as

3D skeletons. We simultaneously recorded speed and accuracy of participant responses to the arithmetic questions in both single calculation task and dual calculation task. In addition, our system is designed for automatic measurement to minimize labor costs. It employs person detection to verify that the entire body skeleton is properly captured in each frame, ensuring the quality of the collected data.

Using the system described above, we can acquire long-term observations of dual-task performance data from each participant over a period of $\alpha$ months. By analyzing subtle changes in dual-task performance data over time, we can predict future cognitive decline as early as possible. The overtime MMSE scores measured every 6 months, with a duration ranging between 2 and 5 years, are used to estimate the tendency towards cognitive decline, which serves as ground truth for training. MMSE assessments should be conducted at 6-month intervals to reduce practice effects [7].

Based on this protocol, we developed a real-world application for dementia prevention (illustrated in Fig. 3), which was installed in three different elderly centers for long-term cognitive monitoring. In this setting, participants must perform the dual-task test at least once every month. Following each test, the input data are refreshed monthly, replacing the data from the oldest month with the most recent data. As a result, the system can forecast the future risk of cognitive decline on a monthly basis. The time between tests can be reduced to 10 days or 1 week for more frequent monitoring, facilitating early detection of potential signs of future cognitive impairment, thus enabling timely interventions to prevent the onset of dementia. At the same time, the collected dataset, comprising paired long-term behavioral observations (dual-task) and clinical records (overtime MMSE scores), provides valuable insight

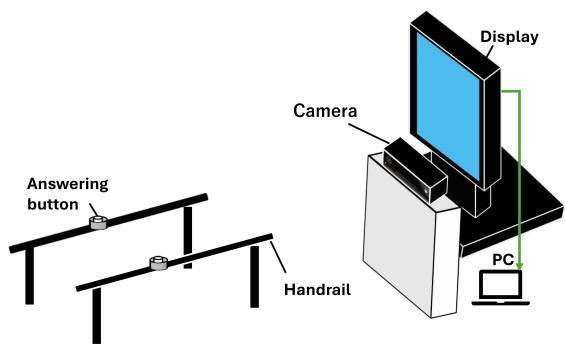

Fig. 2. Dual-task experience system

TABLE I
NINE-DIMENTIONAL BASIC FEATURES USED IN THIS STUDY [17]

| Feature-name | Feature-description |
|---|---|
| Correct-Rate-S | correct answer rate in single calculation task |
| Answer-Time-S | mean answer time in single calculation task |
| Step-Speed-S | mean stepping speed in single stepping task |
| Step-STD-S | standard deviation of stepping speed in single task |
| Correct-Rate-D | correct answer rate in dual calculation task |
| Answer-Time-D | mean answer time in dual calculation task |
| Step-Speed-D | mean stepping speed in dual stepping task |
| Step-STD-D | standard deviation of stepping speed in dual task |
| Thigh-Raise | mean thigh raise during single and dual stepping |

for identifying new precursors of future cognitive changes.

*C. Feature extraction*

In this research, we first calculate basic features such as mean stepping speed, answering speed, correct answer rate, and related metrics from the captured 3D skeletons, alongside answering records for each dual-task measurement. Next, for each sample we record the basic features of all measurements from a given participant in the order of their timestamps within $\alpha$ months. Finally, we extract statistical features using the tstresh [25] algorithm.

Here we adopt the approach proposed by Matsuura et al. [17] to calculate the basic features listed in Table I. Then, the basic features from all dual-task measurements of sample $i$ during an $\alpha$-month period can be organized sequentially as $\widehat{\mathbf{X}}_i = \left\{ \widehat{\mathbf{X}}_i^1, \widehat{\mathbf{X}}_i^2, \dots \widehat{\mathbf{X}}_i^t, \dots \widehat{\mathbf{X}}_i^{N_i} \right\}$, where $\widehat{\mathbf{X}}_i^t \in \mathbb{R}^9$ is calculated from $\mathbf{X}_i^t \in \mathbb{R}^C$ using Matsuura et al.'s method. Here $i$ and $t$ represent the indices of sample and measurement, respectively, $C$ is the number of dimensions of the collected dual-task data from single measurement, and $N_i$ is the number of measurements of sample $i$ during an $\alpha$-month period. These basic features $\widehat{\mathbf{X}}_i^t$ encompass detailed performance metrics from both physical and cognitive tasks in each dual-task measurement. Consequently, it becomes feasible to quantitatively evaluate performance changes during long-term frequent measurements.

The number $N_i$ and frequency of dual-task measurements vary from sample to sample because, in real-world application scenarios, participants conduct the test at their discretion (we establish a minimum frequency of once every month, however, some participants opt for daily assessments, while

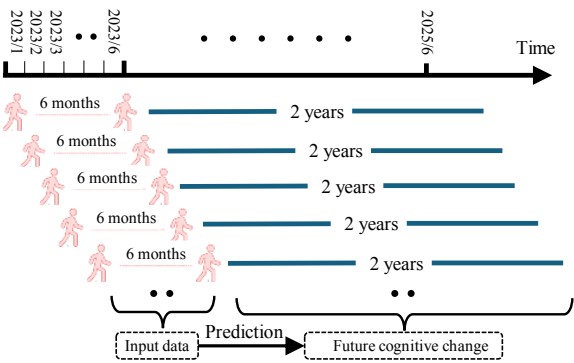

Fig. 3. Cognitive monitoring system based on our approach

others choose to undergo weekly evaluations). Because varying feature durations across samples can pose challenges for machine learning approaches, we employed the tsfresh algorithm, which is capable of extracting statistical features from distributions even in cases of missing data points [25]. The tsfresh algorithm is designed to extract statistical features from time series data, providing comprehensive functionality for calculating statistics such as the number of peaks and lag features. Additionally, it offers the capability to select statistics related to a label, such as the presence or absence of future cognitive decline in this study [25]–[27].

Given $\widehat{\mathbf{X}}_i = \left\{ \widehat{\mathbf{X}}_i^1, \widehat{\mathbf{X}}_i^2, \dots \widehat{\mathbf{X}}_i^t, \dots \widehat{\mathbf{X}}_i^{N_i} \right\}$, where $\widehat{\mathbf{X}}_i^t \in \mathbb{R}^9$ is the input provided to the tsfresh algorithm, and the length of the sequence $N_i$ varies for each sample $i$. The statistical features $\widehat{\mathbf{F}}_i$ are then calculated according to the following expression:

$$\widehat{F}_i = \left\{ f_j(\widehat{X}_i) \right\}_{j=1}^M, \tag{1}$$

where $f_j$ is the $j$-th characterization function related to the data distribution, and $M$ is the number of characterization functions. Here, a characterization function refers to the mathematical operation or transformation applied to time series data to extract relevant features, such as moving averages, Fourier transforms, wavelet transforms, and so on. When implementing the above transformation, the basic features of long-term observations of dual-task performance data $\widehat{\mathbf{X}}_i \in \mathbb{R}^{9 \times N_i}$ for sample $i$ are transformed into $\widehat{F}_i \in \mathbb{R}^S$, where $S$ represents the dimension of the output feature from the tsfresh algorithm, which is common across all samples. While Eq. (1) generates $M$ features, we further refine the feature set through feature selection based on supervised learning, ensuring that $S \leq M$ and retaining only the most effective features. We emphasize that feature selection is conducted solely on the training data. During prediction, the selected feature names identified during training are used for filtering.

*D. Prediction of future cognitive decline*

In this subsection, we begin by clarifying the definition of the label $y_i \in \{0, 1\}$, indicating whether cognitive decline occurs over a future $\beta$-year period, used as the ground truth for training our model. The labeling procedure is fundamental for assessing the efficacy of the proposed approach, and for

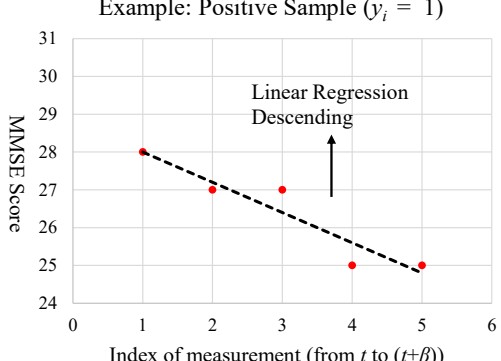

Fig. 4. Linear Regression for positive sample

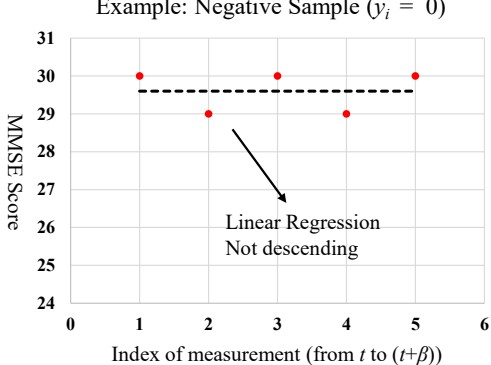

Fig. 5. Linear Regression for Negative example

understanding the significance of this study. Next, we will introduce the machine learning model designed to predict future cognitive decline.

For labeling, we adopt a method based on the rate of change in overtime MMSE scores [2] for each participant associated with the samples used in this study. Let $\mathbf{M}_t^p$ denote the MMSE score for participant $p$ at time $t$. The set of all MMSE scores measured overtime for participant $p$ can then be defined as:

$$\mathbf{M}_D^p = \{\mathbf{M}_1^p, \mathbf{M}_2^p, \ldots \mathbf{M}_t^p, \ldots \mathbf{M}_T^p\}. \qquad (2)$$

We can choose any $t$ with $(t + \beta) \leq T$ as the baseline time. Consequently, long-term observations of dual-task performance data can be defined using all dual-task measurements during the period $[(t - \alpha \text{ months}), t]$. Finally, we can generate one sample for each time $t$ of participant $p$. At the same time, we can extract all MMSE scores $\mathbf{M}_\tau^p = \left\{\mathbf{M}_t^p, \mathbf{M}_{t+0.5}^p, \ldots, \mathbf{M}_{t+\beta-0.5}^p, \mathbf{M}_{t+\beta}^p\right\}$ during the period $[t, (t + \beta) \text{ years}]$. As mentioned previously, MMSE scores are measured every six months. A simple way to detect whether cognitive decline has occurred is to compare MMSE scores at the start and end time points. However, bias inherent in the MMSE itself may influence the accuracy of the prediction. To enhance robustness against MMSE bias (MMSE scores may be either overestimated or underestimated depending on the rater's discretion [7]), we implement linear regression for $\mathbf{M}_\tau^p$ to obtain the rate of cognitive change $\gamma$ (points per year). If $\gamma$ is less than a threshold $\epsilon$ (points

per year), we set the label as positive; otherwise, we set it as negative. Figs. 4 and 5 show the positive and negative examples after linear regression, respectively. In Fig. 4, the rate of cognitive change $\gamma$ is about $-1.5$, which is lower than the threshold ($\epsilon = -0.6$), we regard this sample as a positive one. In contrast, the rate of cognitive change $\gamma$ is larger than the threshold, we regard it as a negative sample in Fig. 5.

The choice of threshold $\epsilon = -0.6$ is based on previous research, which predicts the conversion from mild cognitive impairment (MCI) to dementia with an accuracy of 82%. This value of threshold was chosen as default value because it serves as a clinically useful indicator of cognitive decline [31]. However it can be adjusted flexibly to fit different objectives. For example, setting it lower than -0.6 points per year would detect slight cognitive decline, while setting it higher than -0.6 points per year would detect severe cognitive decline.

After determining the label for each sample, we can train a machine learning model using the feature $\widehat{F}_i$ calculated in Eq. (1) as the input, and the assigned label as the output. Here we use LightGBM, an enhanced version of gradient boosting decision trees that leverages histogram-based features [28]–[30]. LightGBM efficiently completes the optimization process by excluding non-important instances and bundling exclusive features, a strategy that has proven particularly effective in scenarios involving small-sized databases [28]. Hence, we used LightGBM in our study to predict the presence of future cognitive decline.

After training LightGBM model, the prediction of cognitive decline can be computed by the trained model as follows:

$$\widehat{y}_i^{\,l+1} = \widehat{y}_i^{\,l} + \lambda \widehat{h}^{\,l}(\widehat{F}_i), \qquad (3)$$

where $i$ is sample index, and $l$ is the index of iteration in the LightGBM optimization process. Here $\widehat{h}^{\,l}$ represents the output of the regression trees at iteration $l$.

## IV. EXPERIMENTS

### A. Experimental settings

In this section, we describe the dataset utilized in this research and the experimental settings required to validate the effectiveness of the proposed method.

Our data were collected from three different elderly facilities where the dual-task system was installed to enable frequent measurements for long-term cognitive monitoring. MMSE scores required for labeling were measured every six months by professional staff at the same facilities. Since the purpose of our research is to detect early stages before conditions like Alzheimer's disease develop, we excluded samples with baseline MMSE scores (MMSE scores of the first measurement) below 24 points to focus solely on individuals initially with mild cognitive impairment or those who are healthy. However, MMSE scores below 24 points from the second measurement were retained for linear regression in determining labels. Then we had access to a total of 100 samples from 39 participants. A given participant could be associated with multiple samples, because participants were

| | Minimum | Maximum | Median | Mean | STD |
|---|---|---|---|---|---|
| Age | 68 | 96 | 86 | 85.8 | 5.6 |
| MMSE | 15 | 30 | 27 | 26.3 | 3.3 |
| Rate of change points per year | -4.38 | 2.23 | -0.39 | -0.71 | 1.37 |
| No. of tests | 18 | 150 | 39 | 54.5 | 36.9 |

TABLE III
COMPARISON WITH RELATED WORK USING MRI DATA

| Methods | Acc | Sens | Spec | Sens + Spec |
|---|---|---|---|---|
| Peer work | 0.71 | 0.71 | 0.70 | 1.41 |
| Our method | 0.71 | 0.65 | 0.75 | 1.41 |

TABLE IV
COMPARISON RESULTS WITH DIFFERENT DUAL-TASK DURATIONS AND
LABELING THRESHOLDS

| $\alpha$ value | metrics | Labelling threshold $\epsilon$ (points/year) | | | | | |
|---|---|---|---|---|---|---|---|
| | | 0.0 | -0.2 | -0.4 | -0.6 | -0.8 | -1.0 |
| six months | Acc | 0.79 | 0.76 | 0.71 | 0.71 | 0.72 | 0.74 |
| | Sens | 0.92 | 0.88 | 0.74 | 0.65 | 0.62 | 0.60 |
| | Spec | 0.57 | 0.59 | 0.68 | 0.75 | 0.79 | 0.82 |
| | **Sens+Spec** | **1.49** | **1.47** | **1.42** | **1.41** | **1.40** | **1.42** |
| single time | Acc | 0.71 | 0.73 | 0.66 | 0.64 | 0.64 | 0.69 |
| | Sens | 0.79 | 0.80 | 0.69 | 0.57 | 0.41 | 0.39 |
| | Spec | 0.59 | 0.63 | 0.63 | 0.69 | 0.76 | 0.82 |
| | Sens+Spec | 1.38 | 1.43 | 1.32 | 1.26 | 1.16 | 1.21 |

monitored over many years, encompassing multiple cycles ($\alpha$ months + $\beta$ years).

Table II presents the basic characteristics of the samples used in this study. This study was approved by the Research Ethics Committee, Institute of Scientific and Industrial Research, Osaka University (Approval Number: R6-01), and informed consent was obtained from all participants prior to data collection.

The proposed method addresses a binary classification problem. We therefore used accuracy, sensitivity, and specificity (defined by the following equations) as evaluation metrics. We carried out training and evaluation using a cross-validation technique called Leave-One-Group-Out Cross-Validation (LO-GOCV), and computed evaluation metrics accordingly. LO-GOCV assigns samples from one participant to testing, and uses the remaining samples for training.

$$\text{Acc} = \frac{\text{TP} + \text{TN}}{\text{TP} + \text{TN} + \text{FP} + \text{FN}}. \quad (4)$$

$$\text{Sens} = \frac{\text{TP}}{\text{TP} + \text{FN}}, \quad \text{Spec} = \frac{\text{TN}}{\text{TN} + \text{FP}}. \quad (5)$$

In Eqs. (4) and (5), TP, FP, TN, and FN represent the numbers of true positive, false positive, true negative, and false negative, respectively. The parameters $\alpha$ and $\beta$, which denote the periods for dual-task measurements and future cognitive decline prediction, are set as 6 months and 2 years, respectively.

*B. Comparison experiment with peer study*

In this experiment, we evaluate the performance of our model using default threshold $\epsilon = -0.6$ for labeling. Since there is no existing research that uses dual-task data to predict future cognitive decline, we compare our method with a well-established study that uses MRI for this purpose [31]. In this peer study, cognitive decline prediction is performed using a single MRI scan measured at baseline time $T_{\text{baseline}}$. The definition of cognitive decline is similar to ours, determined by the rate of change in MMSE scores over time. However, there is a significant difference in dataset sizes: the peer study relied on data from 698 individuals obtained from ADNI [32].

Table III presents the results obtained by the peer study, and the classification results obtained using the proposed method

with our database. The proposed algorithm achieves a level of performance comparable to the peer study that used MRI images, however, it is superior for the following three reasons:
(1) the cost of implementing our system is much lower than the cost associated with MRI scanning;
(2) Our system is better suited for early detection as it can be deployed in numerous elderly care centers for daily measurements. In contrast, MRI is impractical for early-stage detection due to the challenges of conducting frequent scans, especially when symptoms of cognitive impairment are not yet evident. MRI scans are typically performed only when symptoms appear and medical attention is sought, often indicating that the disease has already progressed to a more advanced stage. Therefore, from this perspective, our proposed approach is more suitable for dementia prevention than the peer study.
(3) the peer study relied on a database that is approximately seven times larger than ours, implying that the performance of our approach would be higher if we used a database size similar to that used in the peer study.

*C. Comparison experiment for sensitivity analysis*

In this experiment, we implement comparisons with different values of the dual-task period $\alpha$ and labelling threshold $\epsilon$. Firstly, we compare the performance of models using a one-time measurement of dual-task performance data with models using multiple dual-task measurements over a 6-month period. The two compared models use the same labeling rule; the only difference being whether long-term observations of dual-task performance data are used as input. During comparison, we evaluate the performance of the two models using not only the default labeling threshold of $-0.6$ but also thresholds ranging from 0.0 to -1.0. Table IV presents detailed results of the performance comparison. Table IV shows that the overall performance of the model with an extended monitoring period (6 months) outperforms that of the model with a one-time dual-task measurement for all labeling threshold values. This result is consistent with our initial analysis, which indicates that long-term observations of dual-task performance data contain clues related to future cognitive decline. When comparing performance across different values of the labeling threshold, we found that $\epsilon > -0.4$ yields slightly better results than $\epsilon \leq -0.4$. This finding, which results from an issue with data imbalance, will be discussed in the next section.

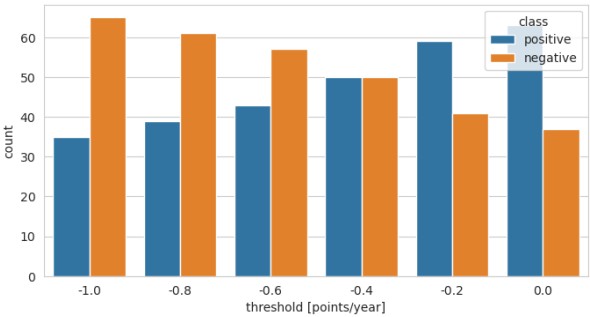

Fig. 6. Numbers of positive and negative samples as a function of different values for the labeling threshold

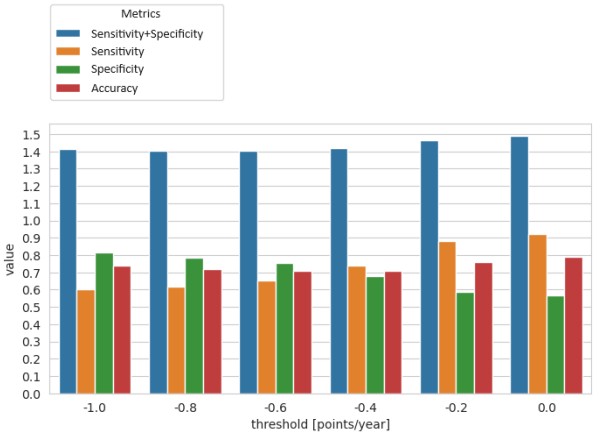

Fig. 7. Classification performance as a function of different values of the labeling threshold

## V. DISCUSSION

In this section, we consider relevant limitations of our study and explore potential solutions to address those limitations. There are primarily two limitations of our research, discussed separately below.

### A. Data imbalance issue

To illustrate the influence of data imbalance more clearly, we investigate the ratio of positive and negative samples for cases with different values of the labeling threshold (Fig. 6). Fig. 7 shows classification performance for different values of the labeling threshold.

Because the number of people with severe cognitive decline (involving 1-point decline per year, as shown on the left side of Fig. 6) is much smaller than the number of those with mild cognitive decline or no decline (involving a 0.0-point decline per year, as shown on the right side of Fig. 6), the number of positive samples for training becomes insufficient when the threshold is set to $-1.0$ ($\epsilon = -0.1$ means that samples with an MMSE score change rate lower than $-1.0$ will be regarded as positive).

This reduction in the number of positive samples results in a degradation of sensitivity. Fig. 7 demonstrates that the

best accuracy, sensitivity, and specificity are achieved when the ratio of positive to negative samples is approximately 2:1, with the labeling threshold set to $0.0$ ($\epsilon = 0.0$ means that samples with an MMSE score change rate lower than $0.0$ will be regarded as positive).

To address the sensitivity degradation caused by data imbalance, a promising solution is to design a weighted loss function that balances contributions from both positive and negative training samples. We will investigate the design of this type of loss function in future work.

### B. Labeling uncertainty issue

Another limitation of this study is the uncertainty in data labeling. This uncertainty stems from the possible measurement errors in MMSE [7] and the possibility of non-linear changes in cognitive function. Although we used simple linear regression to estimate the slope (the change rate per year of the overtime MMSE scores), we did not account for uncertainty in the regression procedure, which could lead to incorrect labeling. These incorrect labels may negatively impact model performance and the accuracy of its evaluation. There are several solutions for this problem, such as employing soft labels during training and excluding samples with high uncertainty. Here, we propose a novel idea related to quantifying and minimizing this kind of uncertainty.

Given $n$ measurements of MMSE taken over the labeling period, $m_i$ denotes the MMSE score at the $i$-th measurement, $\hat{m}_i$ represents the predicted score of the MMSE at the $i$th measurement, $x_i$ denotes measurement date of the $i$-th measurement, and $\bar{x}$ is the average value of the measurement dates. Labelling uncertainty can then be calculated as follows:

$$\sigma_{\text{slope}} = \sqrt{\frac{\frac{1}{n-2}\sum_{i=1}^{n}(m_i - \hat{m}_i)^2}{\sum_{i=1}^{n}(x_i - \bar{x})^2}}. \tag{6}$$

When this uncertainty is small, labeling becomes more reliable. When this uncertainty is large, the likelihood of labeling errors increases, resulting in low-quality labels. Therefore, one potential solution is to use a machine learning model for MMSE change regression instead of linear regression and to minimize this uncertainty during the training of the model. Because this procedure involves the design of an alternative approach diverging from the main paradigm proposed in this study, we will explore it in future work.

### C. Dataset size issue

Our dataset includes 100 samples from 39 subjects, which is smaller than the dataset used in the MRI-based study [31]. Collecting data for long-term observation is challenging, particularly when persuading elderly individuals to participate over extended periods. However, our database is comparable to those used in other dual-task-based studies [8], [9], [11], [12]. To address the issue of data insufficiency, We utilized statistical features extracted by Tsfresh, which represents features from a distribution rather than from a single sample, and employed leave-one-out cross-validation (LOOCV) for evaluation to mitigate bias from the data.

## VI. Conclusion

In this study, we proposed an approach for predicting cognitive decline over the two years following baseline time, based on changes in dual-task observations over a period of six months preceding baseline time. In the proposed method, long-term observations of dual-task performance data are represented as a multivariate time series, and feature selection was performed using the tsfresh [25]. The selected features were used as input to the LightGBM machine learning model [28]. When trained, this model classified data as indicating the presence or absence of future cognitive decline.

The experimental results have shown that the performance of the proposed model is equivalent to the performance reported by a well-known MRI-based study [31]. However, the proposed approach outperforms the MRI-based method when considering the cost of examination and the impact of early-stage detection of cognitive decline. This study represents the first attempt at exploiting long-term behavioral cues for predicting future cognitive decline. The comparison results between a model using a one-time measurement of dual-task performance on the one hand, and a model using multiple measurements over a 6-month period on the other hand, validate the effectiveness of long-term observations of dual-task performance. This research facilitates the regular use of dual-task measurements in real world, for monitoring cognition and the early-stage detection of future cognitive decline.

The primary challenges for future work involve addressing the uncertainty mentioned in the Discussion section, as well as expanding the dataset and refining the loss function to solve the data imbalance problem. Moving forward, our objective is to develop a framework for predicting changes in future cognitive function with high precision, using regular measurements of dual-task performance.

## Acknowledgment

We extend our gratitude to Social Welfare Corporation Misasagikai for their cooperation in data collection. We also extend our gratitude to Prof. Ikeda, Dr. Satake, and Dr. Taomoto in Osaka University Hospital for providing guidance on MMSE scoring.

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
