# OpenReview forum: "Predicting Future Cognitive Decline from Long-term Observations of Dual-task Performance Data"
_IEEE.org/EMBS/BHI/2024/Conference — IEEE BHI'24_

### Official Review · Reviewer_pKQm · 2024-08-03
**Intriguing Method for Cost-Effective Dual-Task Testing in Predicting Cognitive Decline, with Notable Investigable Limitation**

**Overall Rating:** 7
**Confidence:** 2

**Other Quality Metrics:**

(a): excellent
(b): great
(c): great
(d): great

**Questions For The Authors:**

How do you plan to address the potential limitations of using a smaller database compared to the peer study, and do you have any strategies for validating your method on larger, more diverse datasets to support the claims of improved performance further?

**Strengths:**

-	The proposed method is suitable for frequent and routine monitoring, making it highly applicable in real-world settings for early detection and prevention of cognitive impairment.
-	Well-organized and clearly presented methodology

**Summary Of The Paper:**

The paper presents an approach for predicting future cognitive decline by leveraging long-term observations of dual-task performance data. The study demonstrates that changes in dual-task performance over six months can be used to predict cognitive decline over the next two years, achieving accuracy comparable to MRI-based methods. The proposed method offers an alternative to traditional approaches, making it suitable for frequent, routine monitoring in real-world settings. The research also highlights the potential for early detection of cognitive impairment, which is crucial for timely intervention and prevention of dementia.

**Weaknesses:**

The study’s findings are based on a relatively small and specific dataset from three elderly care facilities, which may limit the generalizability of the results to broader or more diverse populations.

---

### Official Review · Reviewer_bQKV · 2024-08-15
**Predicting Future Cognitive Decline from Long-term Observations of Dual-task Performance Data**

**Overall Rating:** 8
**Confidence:** 4

**Other Quality Metrics:**

(a) Great
(b) Excellent
(c) Excellent
(d) Great

**Questions For The Authors:**

Are there other labels to describe the process of cognitive decline besides MMSE? Why don't you select specific patients such as Alzheimer patients and use their diagnosis result as the label?

**Strengths:**

1. The data collection process collects three types of data: 30-second single calculation, 20-second single stepping and 30-second dual-task. These three types of data are used to extract nine features that include both physiological but also mental reflection.

2. The dataset used is from 39 subjects over 5 years. A long period of study time is necessary to explore cognitive decline.

**Summary Of The Paper:**

This paper introduces a novel approach for early detection of cognitive decline, which could help prevent the progression of dementia, including Alzheimer's disease. The authors focus on the dual-task paradigm—a method where participants simultaneously perform physical and cognitive tasks—to assess cognitive function. The study involved 39 participants, with data collected over approximately five years, providing a robust dataset for analysis. The results showed that changes in dual-task performance over time were strongly associated with future cognitive decline.

**Weaknesses:**

The authors didn't control the state of the participant in Fig 2. If they need to make sure that the participant finishes all the tasks with the same state, they need to have the participant to keep still for a period before the task starts.

---

### Official Review · Reviewer_gFXN · 2024-08-26
**Good writing but limit improvement**

**Overall Rating:** 7
**Confidence:** 4

**Other Quality Metrics:**

(a) Clarity of writing;  Excellent
(b) Clinical Significance; Excellent
(c) Methodological Novelty; Great
(d) Experiments and Results good

**Questions For The Authors:**

1. In Sec III, B. The MMSE assessments are conducted every 6 months, but the dual-task evaluation interval is alpha months. How do you align the ground truth with prediction since the frequency might differ?

2. In Sec III, D. "the definition of label y_i". You mix used labels for prediction and labeling for ground truth. If I get it right, the y_i should be your prediction. And your writing may cause misunderstanding.

3. In Sec IV, A. What's the size of the dataset after you excluded samples below 24 points? This value matters since your original dataset is already very small.

4. In Table IV. There is a typo in the second row. The first "Sens" should be "Acc".

**Strengths:**

The paper's organization and writing are excellent and easy to follow. The evaluation is comprehensive and thoroughly explained.

**Summary Of The Paper:**

This paper is the first attempt at exploiting long-term behavioral cues for predicting future cognitive function. Use ML as cost-efficient early detection of cognitive function.

**Weaknesses:**

However, the detection performance compared to the baseline is subtle. The dataset size is insufficient, as the author discussed, causing insufficient performance.

---

### Decision · Program_Chairs · 2024-09-23

Accept